 

# Venous endothelin modulates responsiveness of cardiac sympathetic axons to arterial semaphorin

Denise M Poltavski[1], Pauline Colombier[2], Jianxin Hu[2], Alicia Duron[1,3], Brian L Black[2], Takako Makita[1,3]*

[1]The Saban Research Institute, Children's Hospital Los Angeles, University of Southern California Keck School of Medicine, Los Angeles, United States; [2]Cardiovascular Research Institute, University of California, San Francisco, San Francisco, United States; [3]Darby Children's Research Institute, Department of Pediatrics, Medical University of South Carolina, Charleston, United States

**Abstract** Developing neurons of the peripheral nervous system reach their targets via cues that support directional growth, a process known as axon guidance. In investigating how sympathetic axons reach the heart in mice, we discovered that a combination of guidance cues are employed in sequence to refine axon outgrowth, a process we term second-order guidance. Specifically, endothelin-1 induces sympathetic neurons expressing the receptor Ednra to project to the vena cavae leading to the heart. Endothelin signaling in turn induces expression of the repulsive receptor Plexin-A4, via induction of the transcription factor MEF2C. In the absence of endothelin or plexin signaling, sympathetic neurons misproject to incorrect competing vascular trajectories (the dorsal aorta and intercostal arteries). The same anatomical and physiological consequences occur in $Ednra^{+/-}$; $Plxna4^{+/-}$ double heterozygotes, genetically confirming functional interaction. Second-order axon guidance therefore multiplexes a smaller number of guidance cues in sequential fashion, allowing precise refinement of axon trajectories.
DOI: https://doi.org/10.7554/eLife.42528.001

*For correspondence:
makita@musc.edu

**Competing interests:** The authors declare that no competing interests exist.

## Introduction

The heart is innervated by sympathetic and parasympathetic divisions of the autonomic nervous system, which regulate cardiovascular response to emotion and stress. Normal autonomic function in the heart relies on a proper innervation pattern and establishment of functional synapses by efferent axons from postganglionic autonomic ganglia, which are activated by preganglionic neurons that convey orders from the central nervous system. Postganglionic sympathetic neurons that innervate the heart mostly originate from the cervicothoracic sympathetic ganglia, also called the stellate ganglia (STG). These neurons target the conduction system, cardiomyocytes and coronary vessels of the heart to control heart rate, contractility and myocardial blood flow, respectively. In humans, excess sympathetic input is associated with lethal tachycardia (*Cao et al., 2000*), whereas decreased sympathetic tone is associated with bradycardia and progressive heart block (*Lambert et al., 2010*). Although the anatomy and physiology of the cardiac sympathetic system is well-understood, the processes that result in establishment of correct circuitry have been less studied.

During embryonic development, sympathetic axons throughout the body traverse blood vessels to reach their end-organs, a coordinated organization called neurovascular congruence. While sympathetic axons follow arteries in virtually all other cases, we discovered that cardiac sympathetic axons from the STGs are unique in following veins to reach the heart (*Manousiouthakis et al., 2014*). Specifically, cardiac sympathetic axons from the right STG descend along the right superior

vena cava to reach and innervate the sinoatrial (SA) node and the lateral surface of the ventricles, while those from the left STG extend along the left superior vena cava, the left sinus horn, the sinus venosus, and project further onto the dorsal surface of the ventricular chamber. The selection of these venous routes is an elegant developmental solution that reflects the anatomical location of the primary cardiac STG targets, which are positioned adjacent to or closest to veins rather than arteries (e.g., the SA node is located at the junction of the right atrium and the superior vena cava). In contrast, a separate subset of STG axons project to the thoracic body wall, the arms, and the dorsal region of the neck, and these follow arteries (specifically, intercostal, subclavian, and vertebral arteries, respectively) to reach these non-cardiac targets.

One mechanism that underlies neurovascular congruence is the secretion of factors from the blood vessel that serve as guidance or growth and survival factors for axonal extension. In cardiac sympathetic innervation, we previously demonstrated that endothelin-1 (Edn1), which is selectively expressed by the endothelium of the cardiac veins, directs axonal growth towards and along these venous routes by the subset of STG neurons (approx. 50%) that express the endothelin receptor Ednra (*Manousiouthakis et al., 2014*). In mice, axons from the STG initiate extension toward the heart at embryonic day (E) 14.5 and have reached the heart by E15.5. The functional role of endothelin signaling was demonstrated in Edn1-Ednra signaling deficient mouse embryos, which showed a significant reduction of cardiac sympathetic axons reaching the SA node and the ventricular myocardium at E15.5. This deficiency of sympathetic innervation persisted into the postnatal period, and as a consequence, mutant mice were substantially compromised in ionotropic response to sympathetic activation (i.e., amphetamine-induced increased heart rate). Edn1-Ednra signaling, therefore, is a critical axon guidance mechanism that controls the development and ultimate function of the cardiac sympathetic nervous system. In the larger sense, all sympathetic nerves must select among a plethora of potentially available intermediate vascular targets. It is assumed that the complex circuitry of the sympathetic nervous system reflects the expression of a number of tropic and trophic factors such as Edn1 by different vascular elements, and a diversity of neurons that express the appropriate corresponding receptors for these agents. This framework could explain the selection of specific venous and specific arterial routes by different subsets of STG neurons, although the molecular features that account for this circuitry are still only superficially understood.

In this study, we investigate a further and previously unrecognized feature of Edn1-Ednra signaling in cardiac sympathetic axon guidance. We demonstrate that venous endothelium-derived Edn1 not only instructs Ednra⁺ sympathetic axon guidance to the heart, but also induces the ability of these axons to repel from competing arterial routes by upregulating the expression of Plexin-A4 in Ednra⁺ STG neurons. Lacking this second layer of guidance, cardiac sympathetic axons misproject to ectopic targets, and as a result, cardiac sympathetic innervation and physiology are both compromised. This analysis reveals that positive axon guidance alone is insufficient to accomplish correct circuitry, and defines a molecular mechanism by which axon attraction and repulsion are coordinately controlled to synergistically accomplish proper innervation.

## Results

### Edn1-Ednra signaling deficient STG axons misproject along arteries

In addition to neurons that project via veins to the heart, different subsets of STG neurons follow arteries to supply sympathetic innervation to muscles, glands and blood vessels of the dorsal part of the neck, forelimbs and upper thoracic body wall. Previously, we demonstrated a significant loss of sympathetic axons that follow the superior vena cavae to reach the heart in the absence of Edn1-Ednra signaling (*Manousiouthakis et al., 2014*). Given the choices of arterial routes in their vicinity, it still remained unclear whether Ednra⁺ STG axons do not project at all, follow other arterial routes, or project in an atypical manner when they are not properly instructed by venous endothelium-derived Edn1. The vena cavae, descending aorta, aortic arch and ductus arteriosus, and the vertebral, subclavian, and intercostal arteries represent the sum of available vascular routes by which STG neurons can project. To address this question, we used tyrosine hydroxylase (Th) immunostaining to map sympathetic axonal projections from the STG in mouse embryonic upper thoraxes. During cardiac sympathetic axonal extension along the superior vena cavae, the STGs are located at the axial level of the lowest cervical vertebra (C7) and are fused with the upper thoracic sympathetic chain

ganglia (*Figure 1a*). The right STG complex stretches from the bifurcation of the brachiocephalic artery to the third thoracic vertebra. Thus, in addition to the heart (primarily the cardiac conduction system), neck and right forelimbs, the right STG neurons supply the upper right chest wall by extending axons laterally but not medially along the right posterior intercostal arteries, which originate from the descending thoracic aorta. Similarly, the left STG complex, which spreads from a position dorsolateral to the aortic arch to the third thoracic column, also extends axons laterally but not

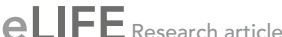

**Figure 1.** Edn1-Ednra signaling mutant STGs project ectopically. (a–h) Wholemount Th immunostaining visualizing sympathetic projections from STGs in E15.5 thoraxes (a–d) and hearts (e–h) from *Ednra*[-/-] (b, f), *Edn1*[-/-] (c, g), *Ece1*[-/-] (d, h) and a control (a, e) embryos. Black arrows denote ectopic medial projections from Edn1-Ednra signaling-deficient STGs to the thoracic aorta (b–d), which is associated with reduced cardiac sympathetic innervations (f, g, h) (*Manousiouthakis et al., 2014*); very rare projections from the STG to the dorsal aorta occur in control embryos (arrow in a). (i) A compiled representation of Th[+] area in the medial upper thorax area in E15.5 endothelin signaling component mutant embryos. For the embryos of each litter, the proportion of Th[+] pixel area within the upper thoracic body wall area (between C7 and T4 vertebrae) between sympathetic chains was measured. Analysis included results of 5 *Ednra* litters (7 controls, 10 *Ednra*[-/-]), 4 *Edn1* litters (9 controls, 9 *Edn1*[-/-]), and 7 *Ece1* litters (18 controls, 15 *Ece1*[-/-]). Error bars; mean ± sd. *p=8.49E-22, **p=8.22E-25, ***p=1.71E-19. (j–q) Serial transverse sections of an E15.5 *Ednra*[-/-] embryo at the levels corresponding to the white dotted arrows in (b) (from the top: T2 vertebral body, the second rib, T3 vertebral body, and the third rib) were immunostained for Th (brown) and counterstained with hematoxylin (blue). (n–q) Magnified views of bracketed areas in j–m. Red arrows point to ectopic medial projections from STG that are associated with thoracic arteries. dAo, descending aorta; es, esophagus; LA, left atrium; lsvc, left superior vena cava; LV, left ventricle; pia, posterior intercostal artery; RA, right atrium; rsvc, right superior vena cava; RV, right ventricle; sv, sinus venosus; stg, stellate ganglion; T, thoracic segment; tr, trachea; X, Xth cranial nerve. Scale bars, 200 μm (a–d), 100 μm (e–h), 200 μm (j–m), 100 μm (n–q).

DOI: https://doi.org/10.7554/eLife.42528.002

medially along the left posterior intercostal arteries to innervate the left upper chest wall. In the presence of normal Edn1-Ednra cardiac sympathetic axon guidance, very few Th[+] axons were found in the dorsal-medial region of the upper thorax (T1-T3 level) (*Figure 1i*), whereas posterior thoracic sympathetic chain ganglia (T4 and below) actively project both medially to the dorsal aorta and laterally along the intercostal arteries.

As we previously reported, mouse mutations in genes encoding Edn1-Ednra signaling components (endothelin receptor *Ednra*, endothelin ligand *Edn1*, and endothelin converting enzyme *Ece1*) all resulted in reduced sympathetic innervation of the heart (*Manousiouthakis et al., 2014*) (*Figure 1f–h*). We found that Edn1-Ednra signaling-deficient STG complexes also abnormally project medially across the dorsal wall of the thoracic cavity, and ultimately reach the descending aorta (*Figure 1b–d and i*). Ectopic projections originating throughout the craniocaudal domain of the STG (i.e., C7-T3 level) were evident from both left and right STG complexes. Histological analyses showed that these ectopic projections were associated with the posterior intercostal arteries and the right subclavian artery, which are visceral branches of the thoracic aorta (*Figure 1j–q*), suggesting that STG axons that either do (*Edn1* and *Ece1* mutants) or should (*Ednra* mutants) express Ednra failed to follow their normal venous routes, and instead take nearby, albeit ectopic, arterial routes to reach improper targets. Arteries are known to secrete a number of tropic and trophic factors (*Brunet et al., 2014*; *Enomoto et al., 2001*; *Francis et al., 1999*; *Honma et al., 2002*; *Makita et al., 2008*); this behavior underlies the arterial selection by most sympathetic axons throughout the body. The observations above indicate that endothelin signaling is required for cardiac sympathetic axons to choose venous routes to the heart and is also required for those axons to not follow ectopic arterial routes.

## Edn1-Ednra signaling deficient STG exhibit reduced repulsive response to arteries in vitro

To further understand the relationship between endothelin signaling and inappropriate STG arterial targets, we explanted STGs in a collagen gel and co-cultured with dissected vascular segments in the absence of any exogenous factors. Without exogenous factors or without coculture with vascular tissue, sympathetic ganglia do not initiate outgrowth. In our previous study, we used this assay to evaluate tropic (attractive) and trophic (growth promoting) effects of venous segments in STG neurite outgrowth and to demonstrate that venous segment-derived Edn1 acts as an attractive factor for Ednra[+] STG neurons in vitro (*Manousiouthakis et al., 2014*). Here, we co-cultured STG explants with thoracic aortic segments, both isolated from embryos at E14.5, a time at which normal STG neurons have already been exposed to venous-derived Edn1. Similar to the response to venous segments, wild-type thoracic aortic segments exhibited a strong attractive effect on wild-type STG neurite outgrowth (*Figure 2a and c*). However, unlike the response to the venous segments, a large fraction of control neurites in the proximal quadrant either stopped midway to the aortic segment (*Figure 2a* yellow dot, 2d) or turned away from the aortic segment (*Figure 2a* red dots, 2d). This is consistent with the presence of chemorepulsive signals originating from the aortic tissue that are received by a subset of STG neurons. Control STGs showed the same behavior when co-cultured with *Ednra*[-/-] aortic segments (*Figure 2c and d*). *Ednra*-deficient STGs showed initial neurite outgrowth towards aortic segments similar to the controls (*Figure 2b and c*). Interestingly, however, a much greater fraction of *Ednra*-deficient neurites grew into or turned towards the aortic segments (*Figure 2b* blue dots; *Figure 2d* blue bars), and reciprocally, fewer neurites stopped or turned away from the aortic segment (*Figure 2b and d*). These observations collectively suggest that, in addition to tropic and trophic factors, the thoracic aorta also expresses factor(s) that are repulsive for a subset of sympathetic axons of the STGs, and furthermore, that Ednra is intrinsically required in these STG neurons to respond to these repulsive guidance cue(s) emanating from the thoracic aorta. The approximately half of control STG neurons that project to the aortic segment (*Figure 2d*) is presumed to overlap with the approximately half of STG neurons that do not express Ednra (*Manousiouthakis et al., 2014*), and therefore are nonresponsive to these repulsive cues.

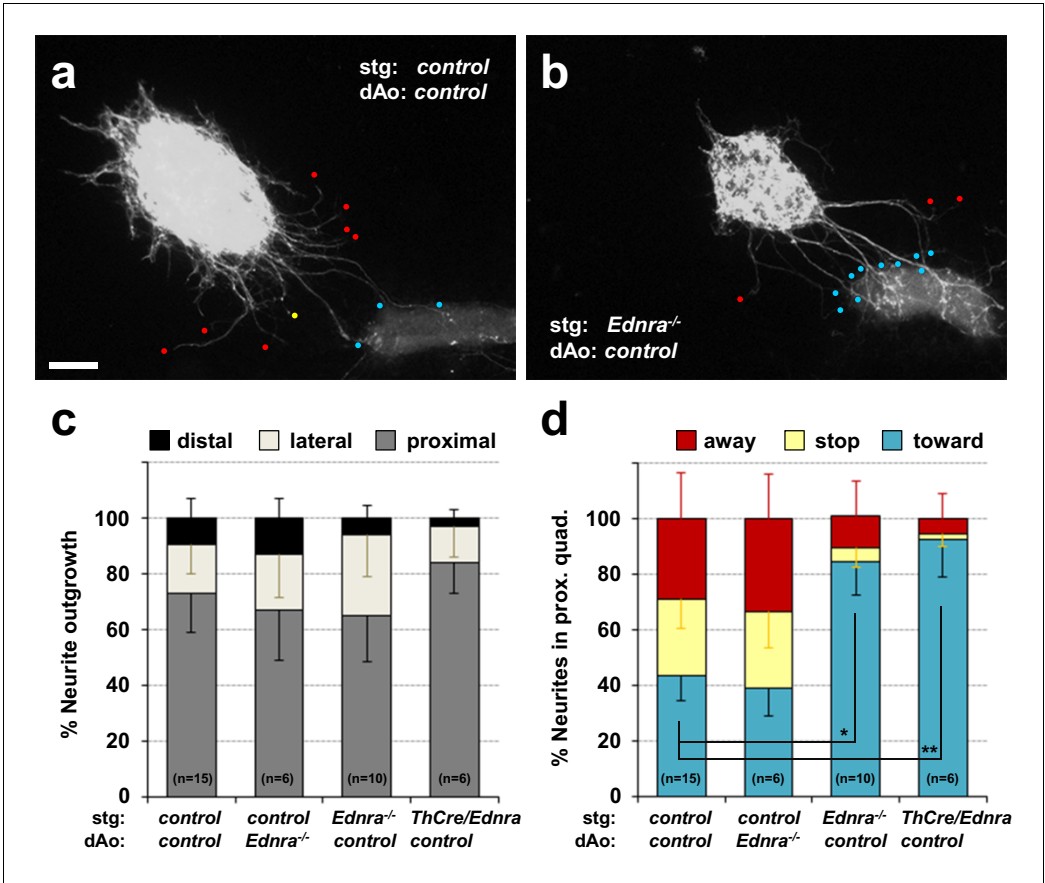

**Figure 2.** Lack of repulsive response to arterial segments in *Ednra*-deficient STGs in vitro. (**a–b**) Neurite outgrowths from E14.5 control STG (**a**) and *Ednra*$^{-/-}$ STG (**b**) cocultured with thoracic descending aorta segments (dAo) dissected from littermate control embryos were visualized by wholemount Th immunostaining. Red, yellow and blue dots denote neurites that have turned away from, have stopped extending toward, or have grown into the vascular segment (proximal quadrant), respectively. (**c**) A compiled representation of directional outgrowth (pixel area in the proximal quadrant) from the STG in terms of relative neurite outgrowth projecting proximally, laterally, or distally to the dAo. For each experiment, total neurite outgrowth from the STG (sum of all four quadrants) was defined as 100%, and neurite outgrowth in each quadrant is shown as a fraction of total outgrowth. (**d**) A compiled representation of growth tip directions of neurites within the proximal quadrant shown in (**c**). Total number of growth tips scored in each proximal quadrant was defined as 100%, and the number of growth tips directing away, stopped, or growing toward the co-cultured dAo segments are shown as a fraction of total neurites in the proximal quadrant. (**c–d**) Analysis includes results of *control:control* (n = 15), *control:Ednra*$^{-/-}$ (n = 6), *Ednra*$^{-/-}$:*control* (n = 10) and *Th-Cre/Ednra:control* (n = 6) pairs from seven different litters. A total of 183 growth tips in the proximal quadrants were scored from the analysis and compiled in **d**). Error bars; mean ±s.e.m.; *p=1.53E-08, **p=7.67E-08. Scale bar, 50 µm (**a–b**).
DOI: https://doi.org/10.7554/eLife.42528.003

## Reduced expression of Plxna4 in endothelin signaling deficient STG neurons

Semaphorin-plexin/neuropilin signaling is known to serve as a repulsive guidance mechanism for developing sympathetic neurons. Plexin-A3 (*Plxna3*) and Plexin-A4 (*Plxna4*) are expressed in sympathetic neurons (*Cheng et al., 2001*), and *Plxna4* mutant mice display ectopic medial projections from the sympathetic ganglia (*Waimey et al., 2008*). *Sema3a* mutation, and neural crest-specific deletion of *Nrp1* (which includes all sympathetic neurons), were also shown to result in ectopic medial projections from the thoracic sympathetic ganglia, coupled with reduced sympathetic innervation of the heart (*Ieda et al., 2007*; *Maden et al., 2012*). The apparent similarity of these ectopic projection phenotypes with those that we observed in endothelin mutants suggested the possibility

that endothelin signaling might interact with semaphorin signaling to control the repulsive response of STG axons to aortic arterial segments.

To ask if endothelin signaling alters the expression of semaphorin receptors, we isolated STG individually from *Ednra*[-/-], *Edn1*[-/-], *Ece1*[-/-] and their littermate control embryos and extracted RNA for real-time PCR analysis of *Nrp1*, *Plxna3*, and *Plxna4* gene expression (*Figure 3a*). In all Edn1-Ednra signaling deficient STGs, *Plxna4* expression was reduced by half, whereas expression of *Nrp1* and *Plxna3* was unaffected. Because the *Ednra* gene is expressed only in a subset (approx. half) of STG neurons (*Manousiouthakis et al., 2014*), the 50% reduction in *Plxna4* expression in *Ednra*-deficient STG could reflect a complete loss of expression in the Ednra[+] population, with the gene still expressed in the other half of STG neurons. To address this possibility, we examined Plexin-A4 protein expression by immunofluorescence detection in histological sections. Indeed, in control embryos, Plexin-A4 was detected in Th[+] cardiac sympathetic axons that descended along the superior vena cavae (*Figure 3d–e,h–i*). In littermate *Ednra* mutant embryos, there was no detectable level of Plexin-A4 expression in sympathetic axons that misprojected along the thoracic aorta and the intercostal arteries (*Figure 3j–k and l–m*, yellow arrows), whereas axons associated with the superior vena cavae retained Plexin-A4 expression (*Figure 3j–k and l–m*, white arrowheads). The former are presumably STG-derived that would normally have projected to the heart; the latter are endothelin-insensitive and presumably represent Ednra-negative subsets of neurons that reach the heart from the STG (as in *Figure 1f–h*). These results suggest that venous Edn1 is an upstream regulator of *Plxna4* gene expression within Ednra[+] cardiac STG neurons.

## The *Plxna4* gene contains an endothelin-induced, MEF2C-dependent transcriptional enhancer

Endothelins regulate gene expression by activation of endothelin-dependent transcription factor binding to conserved elements in several different in vitro and in vivo contexts (*Charité et al., 2001*; *Hu et al., 2015*; *Kawamura et al., 2004*; *Sugimoto et al., 2001*). We previously demonstrated in vivo that myocyte enhance factor *Mef2c* gene expression in the neural crest lineage is regulated by endothelin (*Hu et al., 2015*). Specifically, the *Mef2c* gene contains an endothelin-dependent MEF2C autoregulatory transcriptional enhancer (*Mef2c-F1*) element (*Hu et al., 2015*). Analysis of *Mef2c-F1-lacZ* transgenic embryos showed that the *Mef2c-F1* enhancer is active in sympathetic chain ganglia at E10.5 (*Agarwal et al., 2011*). Furthermore, E9.5 *Mef2c-F1-lacZ* embryos cultured in the presence of Edn1 peptide exhibited robust induction of β-galactosidase activity within trunk neural crest cell derivatives (*Hu et al., 2015*). These observations collectively imply that MEF2C in developing sympathetic neurons could regulate *Plxna4* expression in response to endothelin signaling. To further understand how MEF2C might regulate *Plxna4* expression in response to endothelin signaling activation, we analyzed MEF2 binding peaks from cortical neurons from a previously published MEF2 ChIP-seq dataset (*Telese et al., 2015*), and we also assessed evolutionary conservation between mouse, human, and opossum in the *Plxna4* locus using the VISTA enhancer browser (*Frazer et al., 2004*). Using these two complementary approaches, we found a highly conserved MEF2 binding site ~50 kb upstream of the transcription start site (TSS) and four highly conserved MEF2 binding sites located in the third intron of *Plxna4* gene (*Figure 4a*). Each of these five MEF2 sites corresponded to regions identified in ChIP-seq as bona fide sites of MEF2C binding (*Telese et al., 2015*).

To test whether the *Plxna4* gene was responsive to endothelin signaling via MEF2C, we analyzed RNA expression in FACS sorted neural crest cells purified from *Mef2c* neural crest-specific knockout (*Mef2c*[flox/-]; *Wnt1a-Cre*) [*Mef2c*[NCKO]] and littermate control embryos cultured in the presence or absence of exogenous Edn1. To minimize detection of endothelin-independent induction of *Plxna4* gene expression, we treated explanted E9.5 embryos, a time at which *Plxna4* is not yet actively expressed in sympathetic progenitor cells. Importantly, we observed robust upregulation of *Plxna4* gene expression in Edn1-treated control neural crest-derived cells (*Figure 4b*). In the absence of *Mef2c* function in the neural crest, Edn1 induction of *Plxna4* expression was significantly attenuated compared to Edn1 induction of *Plxna4* expression in control neural crest (*Figure 4b*), supporting the notion that *Plxna4* is a MEF2C target in response to endothelin signaling.

If MEF2C mediates *Plxna4* expression in an endothelin-dependent manner in vivo, a prediction is that absence of *Mef2c* should results in misprojection of cardiac sympathetic axons from the STG, as seen in *Plxna4* (*Waimey et al., 2008*) and Edn-Ednra signaling component mutant embryos (*Figure 1b–d*). To address this, we used wholemount Th visualization of thoracic sympathetic axons



**Figure 3.** Reduced expression of *Plxna4* in *Ednra*-deficient STGs. (a) Quantitative real-time PCR analysis for class three semaphorin receptor and co-receptor genes. Total RNA was extracted from individual E14.5 STG isolated from total seven different litters, reverse transcribed, and evaluated for gene expression as described in Methods. Normalized averaged absolute values for control STGs (n = 15) were scaled at 1.0. Error bars; mean ±s.e.m.; *p=0.002, **p=0.0016, ***p=0.095. (b–m) Serial histological sections of *Ednra*$^{-/-}$ (c, f–g, j–m) and littermate control (b, d–e, h–i) immunostained for Th (b–c) or immunofluorescence labeled for Th (green) or Plexin-A4 (magenta). Magnified views from an adjacent section of the bracketed areas in b) including the right superior vena cava (rsvc) and left superior vena cava (lsvc) are shown in (d-e and h–i), respectively. Magnified views from an adjacent section of the bracketed areas in (c) including the right superior vena cava (rsvc), left superior vena cava (lsvc), and a posterior intercostal artery (pia) are shown in (f–g), (j–k) and (l-m), respectively. A control pia is not shown because there are no axon misprojections to the pia in normal embryos. White arrowheads point to expression of Plexin-A4 in Th$^+$ cardiac sympathetic axons that are associated with venous routes. Yellow arrows denote lack of Plexin-A4 in Th$^+$ sympathetic axons that are misprojecting along arteries. aAo, ascending aorta; pia, posterior intercostal artery. Scale bars, 200 µm (b–c), 25 µm (d–g), 50 µm (h–k), 50 µm (l–m).
DOI: https://doi.org/10.7554/eLife.42528.004

in E15.5 *Mef2c*-deficient embryos. Indeed, we observed ectopic medial projections from the STG (including T1 through T3 sympathetic chain segments) towards the dorsal aorta in *Mef2c* mutant embryos (*Figure 4c–d*). This misprojection phenotype is strikingly comparable to that observed in Edn1-Ednra signaling component deficient embryos (*Figure 1b–d*). We also evaluated the expression of Plexin-A4 in different subsets of sympathetic axons in control and *Mef2c*$^{NCKO}$ mutant embryos (*Figure 4e–l*). Just as observed in *Ednra* mutants (*Figure 3b–m*), axons associated with the

**Figure 4.** The *Plxna4* locus contains several *bona fide*, conserved MEF2C sites and expression is induced by endothelin signaling in a MEF2C-dependent manner. (a) Schematic diagram of conserved MEF2C binding sites in the *Plxna4* locus. The *Plxna4* locus is shown in the center with transcriptional start site (TSS, bent arrow) and exons (vertical black lines) shown. The track shown above the locus schematic represents a genome browser view of MEF2C binding peaks in the *Plxna4* locus obtained from MEF2 ChIP-seq in E15.5 mouse neurons. The red (upstream) and orange

*Figure 4 continued on next page*

*Figure 4 continued*

(intronic) peaks shown below the locus schematic represent VISTA browser tracks of evolutionary conserved regions in the *Plxna4* locus containing conserved MEF2 binding sites in mouse, human, and opossum. rVista alignment of mouse and human genomic sequences showing a detailed view of the MEF2 binding sequences are indicated below. (b) Relative endothelin-dependent induction of *Plxna4* gene expression in neural crest cells sorted and isolated from *Mef2c*-deficient and control embryo explants treated with Edn1 and measured by RNA-sequencing. Error bars; mean ± s.d.. (c–d) Wholemount Th immunostaining visualizing sympathetic projections from STG in E15.5 thoraxes from *Mef2c^{KO}* (d) and a littermate control (c) embryos. Black arrows denote ectopic medial projections from Mef2C-deficient STGs to the thoracic aorta (d). (e–f) Serial histological sections of *Mef2c^{NCKO}* mutant embryo (f, i–l) and littermate control (e, g–h) immunostained for Th (e–f) or immunofluorescence labeled for Th (green) or Plexin-A4 (magenta) (g–l). Magnified view from an adjacent section of the bracketed areas in (e) including the left superior vena cava (lsvc) is shown in (g–h). Magnified views from an adjacent section of the bracketed areas in (f) including the left superior vena cava (lsvc), and an aortic arch (AA) are shown in (i–j) and (k–l), respectively. White arrowheads point to expression of Plexin-A4 in Th^+ cardiac sympathetic axons that are associated with venous routes. Yellow arrows denote lack of Plexin-A4 in Th^+ sympathetic axons that are associated with arteries. Scale bars, 200 μm (c–d, e–f), 50 μm (g–l).
DOI: https://doi.org/10.7554/eLife.42528.005

superior vena cavae retained Plexin-A4 expression in *Mef2c^{NCKO}* mutant embryos (**Figure 4f and i–j**). In contrast, sympathetic axons associated with the thoracic aorta did not express Plexin-A4 in *Mef2c^{NCKO}* mutant embryos (**Figure 4f and k–l**), which we also observed in *Ednra* mutants (**Figure 3c,j–m**). Taken together, these observations are consistent with the interpretation that in the endothelin-dependent subset of cardiac sympathetic axons, Edn1 induces *Plxna4* expression via MEF2C to support proper axon guidance, whereas endothelin-independent subsets of STG axons express *Plxna4* in a MEF2C-independent manner.

## Genetic relationship between *Ednra* and *Plxna4* in cardiac sympathetic axon guidance

To further determine if the *Ednra* and *Plxna4* genes are functionally related in cardiac sympathetic axon guidance, we took a genetic approach that involved analysis of postnatal mice that were heterozygous at the *Ednra* and *Plxna4* loci. A phenotypic consequence in double heterozygotes when each single heterozygote is normal suggests that the two genes might be functionally related or function in a common pathway. In any case, a genetic interaction observed in compound heterozygotes suggests that lowering the expression of both gene product together reduces downstream signaling below a threshold level where a phenotype emerges.

As we described previously (**Manousiouthakis et al., 2014**), the extent of sympathetic innervation of the heart is compromised in *Th-Cre; Ednra^{fl/-}* conditional null mice (**Figure 5d**, **Figure 5—figure supplement 1a**). We observed a similar degree of impairment in sympathetic-neuron-specific *Th-Cre; Ednra^{fl/+}; Plxna4^{+/-}* double heterozygous mice (**Figure 5c**, **Figure 5—figure supplement 1a–b**). In sections through the SA node, using HCN4 as a marker of conduction system cardiomyocytes, Th^+ nerve fibers were obviously reduced in number and caliber, although still present, in double heterozygous hearts (**Figure 5k**, **Figure 5—figure supplement 1b**). We used synaptophysin (Syp), which is expressed in presynaptic sites, to determine if active synapses were formed by these nerve endings. The level of Syp staining was reduced commensurate with the reduction of Th staining (**Figure 5l–m**, **Figure 5—figure supplement 1b**), although not altered in a manner that would suggest an independent defect in synapse formation. A similar deficiency in sympathetic axon density and presynaptic assembly was evident in the ventricular myocardium (**Figure 5u–v**). Because these residual nerves are present at similar levels in both double heterozygous and conditional *Ednra*-null hearts (**Figure 5n–p and w–x**, **Figure 5—figure supplement 1a–b**), we interpret these to be a separate subset of sympathetic nerves that reach the heart in an endothelin signaling-independent manner.

To more rigorously examine the endothelin-plexin genetic interaction and its functional consequences, we undertook electrocardiography (ECG) analysis. As described previously (**Manousiouthakis et al., 2014**), amphetamine treatment acutely induces noradrenaline release at sympathetic termini. In turn, noradrenalin activates beta-adrenergic receptors in the cardiac conduction system and the resulting increase in heart rate is a quantitative measure of cardiac sympathetic nerve density and functionality. Mice heterozygous at the *Ednra* or *Plxna4* loci individually responded normally to amphetamine in the acute period (<2 min after injection) by increasing heart rate (**Figure 6a**), which indicates that sympathetic innervation and functionality are insensitive to loss

**Figure 5.** *Ednra* and *Plxna4* interact genetically. (**a–d**) Wholemount immunostaining visualizing the dorsal surface of hearts from a 7-week-old *Th-Cre; Ednra*<sup>fl/+</sup>*; Plxna4*<sup>+/-</sup> double heterozygote (**c**), a *Th-Cre; Ednra*<sup>fl/-</sup> mutant (**d**), and single heterozygote controls *Th-Cre; Ednra*<sup>fl/+</sup> (**a**) and *Plxna4*<sup>+/-</sup> (**b**). a littermate control (**a**). (**e–x**) 300 μm vibratome sections immunofluorescence labeled for HCN4 (cyan), Th (magenta) and SYP (yellow) visualizing

*Figure 5 continued on next page*

*Figure 5 continued*

sympathetic presynaptic morphologies in the SA node (**e–p**) and left ventricular myocardium (**q–x**) of an adult mice. ca, coronary (circumflex) artery, ivc, inferior vena cava. Scale bars, 500 μm (**a–d**), 50 μm (**e–x**).

DOI: https://doi.org/10.7554/eLife.42528.006

The following figure supplement is available for figure 5:

**Figure supplement 1.** Quantitative assessment of *Ednra* and *Plxna4* genetic interaction.

DOI: https://doi.org/10.7554/eLife.42528.007

of one copy of either *Ednra* or *Plxna4*. However, *Ednra; Plxna4* double heterozygous mice (*Ednra$^{+/-}$; Plxna4$^{+/-}$*) and conditional *Ednra-Plxna4* double-heterozygous mice (*Th-Cre; Ednra$^{fl/+}$; Plxna4$^{+/-}$*) exhibited a severely blunted acute ionotropic response, nearing the completely eliminated response of homozygous sympathetic-neuron-specific *Ednra* mutants (*Th-Cre; Ednra$^{fl/-}$*) (**Figure 6a**). These observations support the interpretation that the *Ednra* and *Plxna4* genes are coupled in cardiac sympathetic axon guidance in a physiologically relevant manner. Amphetamine eventually induces noradrenaline and adrenaline release from the brain and adrenal glands, respectively, which ultimately influence heart rate through systemic circulation. This delayed response (monitored 10 min after

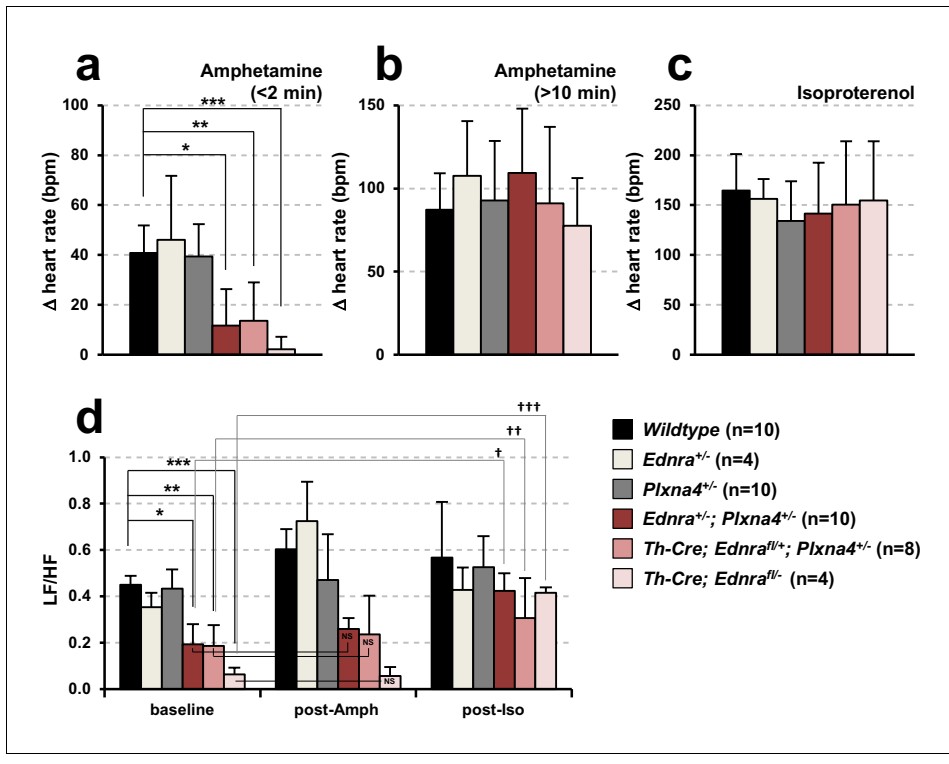

**Figure 6.** *Ednra-Plxna4* double-heterozygous mice exhibit diminished cardiac sympathetic nerve activity. (**a–c**) Changes in heart rate following amphetamine (a;<2 min, (b;>10 min) and isoproterenol (**c**) administration measured in indicated 7-week-old single heterozygotes, double heterozygotes, conditional *Ednra* mutants, and their littermate wildtype controls. Error bars; mean ± s.e.m.; *p=8.95E-05, **p=4.88E-04, ***p=2.55E-05. (**d**) A compiled representation of the low-frequency: high-frequency domain (LF/HF) ratio at the baseline, post-amphetamine and post-isoproterenol administration in mice of the indicated genotypes. Frequency-domain was extracted from power spectral analysis of heart rate variability (HRV) shown in **Figure 6—figure supplement 1**. Error bars; mean ± s.e.m.; *p=0.0001, **p=0.0002, ***p=1.02E-06, †p=0.03, ††p=0.077, †††p=8.39E-05. NS, not significant.

DOI: https://doi.org/10.7554/eLife.42528.008

The following figure supplement is available for figure 6:

**Figure supplement 1.** Reduced baseline sympathetic activity in Ednra-Plxna4 trans-heterozygous heart.

DOI: https://doi.org/10.7554/eLife.42528.009

injection) was fully intact in all double-heterozygous mice (*Figure 6b*), which demonstrates that the noradrenergic and adrenergic systems in the central nervous system and adrenal glands are intact in double heterozygous mice. Furthermore, these mice were normally responsive to the beta-adrenergic receptor agonist isoproterenol (*Figure 6c*), indicating that the cardiomyocyte-specific components of the conduction system and myocardium that respond to sympathetic inputs are functional in the double-heterozygous background.

Sympathetic versus parasympathetic activities can also be evaluated by heart rate variability (HRV) (*Akselrod et al., 1981*). HRV represents physiological rhythm fluctuations between consecutive beats: beat-to-beat fluctuation is minimal in the resting state, whereas in the conscious state, the interval between consecutive beats becomes more variable because of fluctuations in the opposing actions of sympathetic and parasympathetic inputs. HRV is divided into low-frequency (LF) and high-frequency (HF) components, and commonly described by the LF/HF ratio (*Figure 6d*). In *Th-Cre; Ednra$^{fl/-}$* mutants, in which sympathetic innervation and activity are reduced (*Figure 5d, n–p and w–x*), a significant reduction of the LF component was observed (*Figure 6d*, *Figure 6—figure supplement 1c*), which indicates less cardiac sympathetic neural activity in the baseline state. *Th-Cre; Ednra$^{fl/+}$; Plxna4$^{+/-}$* ± also exhibited a similar reduction of the LF/HF ratio (*Figure 6d*, *Figure 6—figure supplement 1b*), coinciding with their reduced cardiac sympathetic innervation (*Figure 5c, k–m and u–v*). Single heterozygote mice for either *Ednra* or *Plxna4* were unaffected (*Figure 6d*). Amphetamine treatment increases the LF/HF ratio in normal mice by activating sympathetic activity, but this did not change in the compound mutants, which coincides with their attenuated change in heart rate after acute amphetamine treatment (*Figure 6d*). Similarly, isoproterenol treatment increased the LF/HF ratio in all mice (*Figure 6d*). These observations collectively demonstrate that endothelin control of *Plxna4* expression is intrinsically required for Ednra$^+$ STG neurons to follow venous routes to reach the heart, and that this subset of STG neurons is required for normal cardiac sympathetic function.

Endothelin signaling is also required for a subset of axons from the superior cervical ganglia (SCG) to properly project along the external carotid arteries to targets in the head and neck (other neurons project along the internal carotid arteries in an endothelin-independent manner to reach different targets in the head) (*Makita et al., 2008*). This system relies on spatial and temporal expression of the ligand Edn3 by smooth muscle cells of the 3$^{rd}$ pharyngeal arch arteries during the E11.5–12.5 period, which is in contrast to expression of Edn1 by endothelium of the vena cavae at E14.5 and onward for STG projections to the heart. Despite the utilization of these two different ligands, the SCG and STG both employ the same endothelin receptor, Ednra, for guidance. *Plxna4* mutants show an SCG misprojection phenotype (*Waimey et al., 2008*) that results in fewer SCG axons extending along the external carotid arteries (*Figure 7c and g*). Because of these observations, we histologically analyzed E14.5 *Ednra$^{+/-}$; Plxna4$^{+/-}$* ± heterozygote embryos for evidence of genetic interaction of the *Ednra* and *Plxna4* genes in SCG axon guidance. Interestingly, however, *Ednra-Plxna4* double heterozygotes were unaffected in SCG external carotid artery projection (*Figure 7b and f*). This indicates that *Ednra-Plxna4* interaction is not a general phenomenon for all sympathetic axon guidance, but rather is specific for the cardiac sympathetic axons from the STG. One possible explanation for this observation is that the *Plxna4* and *Ednra* mutant SCG phenotypes are superficially similar but are not identical: absence of *Plxna4* results in randomized outgrowth (*Figure 7c*, red arrows), whereas absence of *Ednra* eliminates guidance towards the external carotid arteries (*Figure 7d*). Consequently, in SCG axon outgrowth, Ednra and Plxna4 may have different roles that do not synergize in *Ednra$^{+/-}$;Plxna4$^{+/-}$* ± heterozygotes. In contrast, the STG projection and cardiac innervation phenotypes of *Ednra* mutants, *Plxna4* mutants, and *Ednra$^{+/-}$;Plxna4$^{+/-}$* ± heterozygotes are all seemingly identical (*Figure 1*, *Figure 5*).

## Ednra deficient STG fail to repel arterial semaphorin in vitro

The activation of class A plexins (with co-receptor neuropilins) requires interaction with secreted class three semaphorins (*Janssen et al., 2012*). Among class three semaphorins, Sema3a and Sema3f serve as ligands for Plexin-A4 in various biological contexts (*Suto et al., 2005*; *Yaron et al., 2005*). Sema3a and Sema3f are both expressed by the dermomyotome, forelimbs and paraxial mesoderm during the E10.5–12.5 period (*Kawasaki et al., 2002*; *Maden et al., 2012*), and *Sema3a* and *Sema3f* mutant embryos independently display medial ectopic projections from sympathetic ganglia in the upper thorax (*Maden et al., 2012*). We confirmed selective expression of Sema3a and

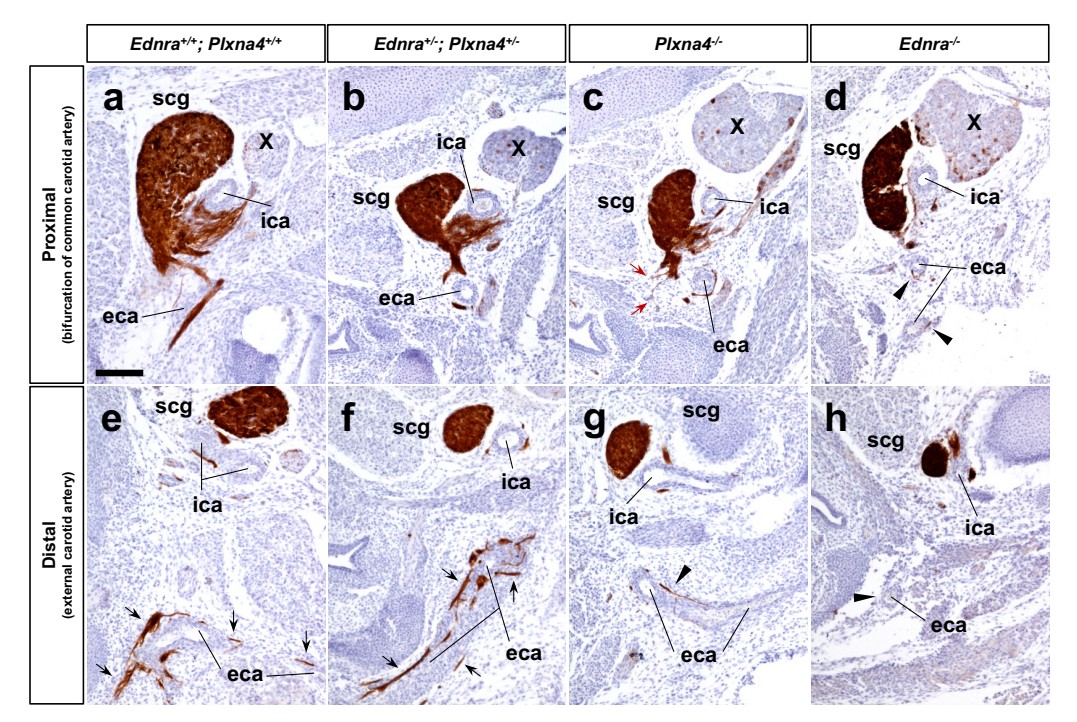

**Figure 7.** Normal SCG axonal projections along the external carotid arteries in Ednra-Plxna4 trans-heterozygotes. (a–h) Transverse sections from E14.5 *Ednra*[+/-]; *Plxna4*[+/-] (b, f), *Plxna4*[-/-] (c, g), *Ednra*[-/-] (d, h) mutant embryos and a littermate control embryo at the level of carotid artery bifurcation (a–d) and the upper larynx (e–h) stained with Th showing sympathetic axons from the SCG extending along the external carotid artery in the trans heterozygote. Black arrows point to normal projections of SCG axons along the external carotid arteries (e, f). Red arrows indicate ectopic (random) sprouts from the SCG in *Plxna4*[-/-] mutant embryo (c). Arrowheads denote reduced or defective SCG projections along the external carotid arteries (d, g–h). eca, external carotid artery; ica, internal carotid artery; scg, superior cervical ganglion. Scale bar, 100 μm (a–h).
DOI: https://doi.org/10.7554/eLife.42528.010

Sema3f in the smooth muscle layer of the upper thoracic aorta during the critical period of cardiac sympathetic axonal extension to the heart (*Figure 8—figure supplement 1a–d and e–h*). To test the hypothesis that arterial semaphorin(s) repel STG axons in an Edn1-Ednra-dependent manner, we next used in vitro STG explant culture with COS cell aggregates. As noted above, in minimal growth medium with no growth/trophic factors added, STG isolated from E14.5 control embryos and explanted alone do not extend neurites (*Manousiouthakis et al., 2014*). Regardless of *Ednra* or *Plxna4* genotype, STG exhibited a robust tropic and trophic response to aggregates of untransfected COS cells, indicating that COS cells secrete factors (which are not defined) that are active on sympathetic neurons (*Figure 8a–d*). When control STGs were placed between untransfected and Sema3a-transfected COS cell aggregates (GFP[+]) (*Figure 8i*), neurites only projected away from the Sema3a-expressing cells (*Figure 8e and j*), demonstrating the repulsive nature of Sema3a signaling on all STG sympathetic axons (all of which, including Ednra[+] neurites in control STGs, express Plexins). Neurites from *Ednra*[+/-] or *Plxna4*[+/-] STGs showed a similar response to Sema3a-expressing COS aggregates as control STGs (*Figure 8j*). In contrast, however, *Ednra*-deficient STGs exhibited a fraction of neurite outgrowth that was attracted towards the Sema3a-expressing COS aggregate (*Figure 8g and j*). A similar response was observed in *Ednra-Plxna4* double heterozygote STG and *Plxna4*-null-STG explants (*Figure 8f, h and j*). As described above, *Ednra* is expressed only by half of STG neurons while *Plxna4* is expressed in most STG neurons.

## Discussion

The complexity of the nervous system is vast. Sympathetic nerves have partially solved the problem of how to reach appropriate targets by restricting their extensions to lie along blood vessels.

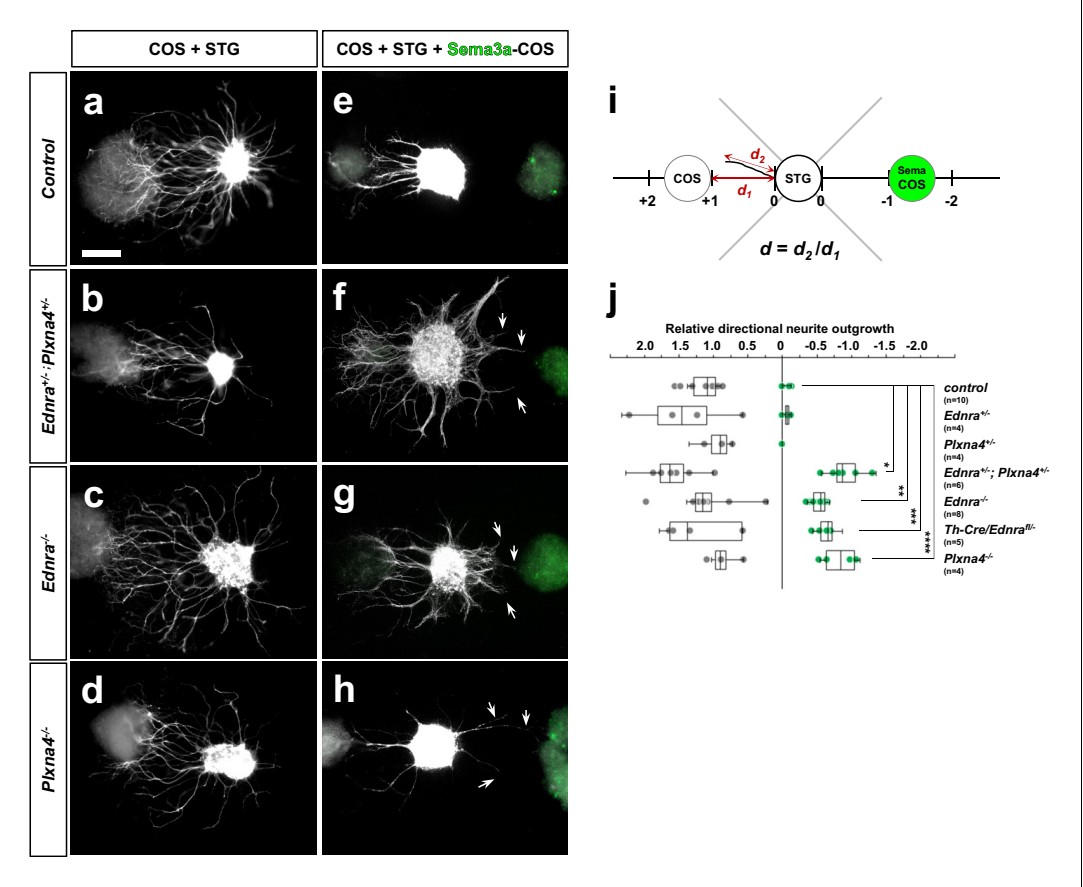

**Figure 8.** *Ednra*-deficient STGs show reduced repulsive response to Sema3a in vitro. (**a–h**) Neurite outgrowth from explanted STG co-cultured with COS cell aggregates were visualized by whole mount Th staining. STG dissected from E14.5 embryos of the genotypes indicated at left were cultured in the proximity of untransfected-COS cell aggregates (**a–d**) or in between untransfected and Ad-GFP-2A-mSema3a- transfected-COS cell aggregates (**e–h**). Sema3a expression of transfected-COS cell aggregates are confirmed by GFP expression (green; **e–h**). The repulsive effect was measured by neurite outgrowth length towards the untransfected-COS aggregate as positive numbers and towards Sema3a-transfected-COS aggregates as negative numbers, as indicated in (**i**). To control for experimental variation, the distance between the STG and the COS aggregates ($d_1$) was defined as 1.0 or −1.0 and each neurite outgrowth length toward either COS aggregate ($d_2$) was normalized as a relative value (**d**). (**j**) A compiled dot-whisker plot representation of the repulsive response of STG to Sema3a. Each dot represents the averaged neurite outgrowth from each STG towards untransfected-COS (grey dots) or Sema3a-transfected-COS (green dots) aggregates in each experimental culture. The number of experimental replicates for each STG genotype is as indicated, and analysis includes the total counts of 98 wildtype, 31 *Ednra*$^{+/-}$, 24 *Plxna4*$^{+/-}$, 91 *Ednra*$^{+/-}$; *Plxna4*$^{+/-}$, 102 *Ednra*$^{-/-}$, 57 *Th-Cre/Ednra* and 51 *Plxna4*$^{-/-}$ STG neurites. *p=5.93E-08, **p=7.12E-10, ***p=3.48E-08, ****p=4.97E-07.

DOI: https://doi.org/10.7554/eLife.42528.011

The following figure supplement is available for figure 8:

**Figure supplement 1.** Semaphorins are expressed in arteries but not veins.
DOI: https://doi.org/10.7554/eLife.42528.012

However, even with this limitation, these nerves must select from among a number of arteries, and as now recognized, also veins, that lie in their vicinity. Selectivity is presumed to be accomplished by expression of a variety of guidance cues by different vascular elements and by expression of a variety of receptors by different subsets of sympathetic neurons. Only a relatively small number of attractive or repulsive guidance cues are known and are too few to explain the complexity of the sympathetic system at least according to a first-order (one-to-one) relationship between guidance cue and guided nerve. One possibility is that many more guidance factors exist but have not yet been discovered. Instead, our study reveals a second order of regulation, in which detection of a first signal (endothelin) by the Ednra$^+$ subset of STG neurons induces expression of a second axon guidance receptor (Plexin-A4) that further refines the selection of vascular targets (*Figure 9*). Second-order

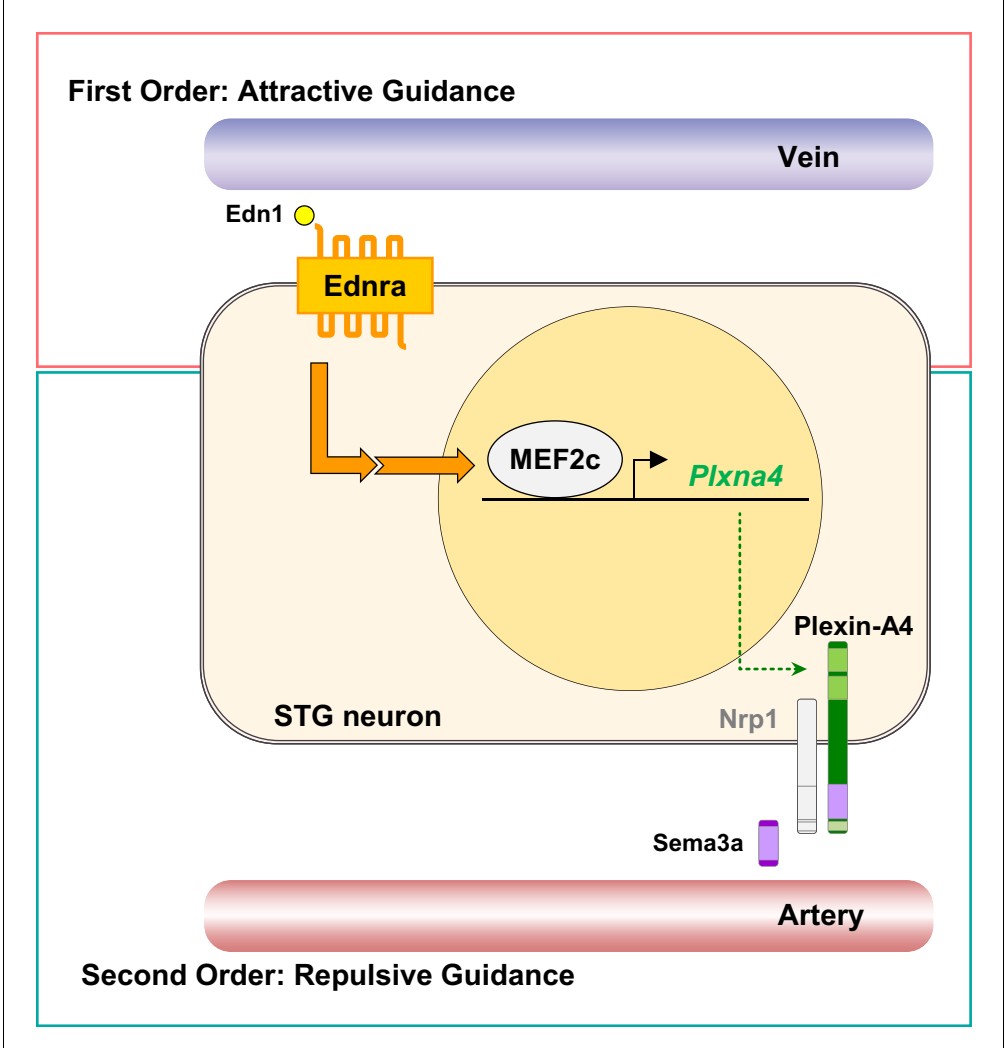

**Figure 9.** A model of second-order axon guidance for cardiac sympathetic axons. An Ednra-expressing STG neuron located in the vicinity of distinct vascular trajectories (vein and artery) sees venous-derived endothelin as a primary attractive guidance cue and initiates axonal extension toward the vein (First Order). Endothelin (Edn1-Ednra) signaling induces MEF2C transcription factor binding to MEF2C response element(s) in the *Plxna4* gene promoter to upregulate expression of *Plxna4* gene in venous-associated Ednra[+] STG neurons. Plexin-A4 then serves as a repulsive guidance receptor for arterial Sema3a (Second Order), which further restricts endothelin-dependent axonal extension along the venous routes by keeping Ednra[+] axons from being attracted toward arterial trajectories.

DOI: https://doi.org/10.7554/eLife.42528.013

guidance allows a smaller number of guidance cues to be deployed combinatorially to support proper peripheral innervation. This novel feature of axon guidance, which we have studied in the heart, might be more generalizable to other aspects of the peripheral nervous system.

Our results demonstrate that second-order guidance for cardiac sympathetic axons involves Sema3-Plexin-A4 signaling. Arterial selective expression of secreted class three semaphorins (*Figure 8—figure supplement 1a–d and e–h*), phenotypic (neuroanatomical and physiological) similarities between Edn1-Ednra signaling mutants and Sema3a-Plexin-A4 signaling mutants, and observations from STG explant neurite outgrowth assays collectively support this model. Plexin-A4 can also be directly activated by membrane-bound class six semaphorins (*Janssen et al., 2010*; *Suto et al., 2005*; *Suto et al., 2007*). We detected Sema6a expression in arterial smooth muscle during the critical time period for cardiac sympathetic axonal projections along vascular routes

(*Figure 8—figure supplement 1i–l*). Sema6a expression was also detected within cardiac sympathetic neurons and axons, whereas Sema6b was not expressed in neurons or vasculature (*Figure 8—figure supplement 1i–l and m–p*). Unlike Sema3a, there is no prior indication that *Sema6a* mutation results in a cardiac sympathetic phenotype, although we have not yet firmly ruled out if Sema6a influences endothelin-dependent repulsive guidance via Plexin-A4.

The importance of Plexin-A4 as a receptor for Sema3 signals in second order guidance of cardiac sympathetic axons is strongly implicated by the misprojection phenotype of *Plxna4* mutants (*Waimey et al., 2008*) and of *Ednra-Plxna4* double heterozygotes (this study). Although mutation of *Plxna3* alone does not result in this phenotype, the *Plxna3* and *Plxna4* double mutant was more severely affected (more ectopic thoracic projections) than *Plxna4* single mutants, implying the possibility of functional redundancy. Plexin mediation of class three semaphorin signaling is thought to require the coreceptor neuropilin, and *Nrp1* mutation also results in severe ectopic medial projections from the sympathetic ganglia (*Kawasaki et al., 2002*). It should be noted that Sema3a-Nrp1 signaling also controls sympathetic progenitor cell migration and then the caudal extension of post-migratory sympathetic neurons during the E9.5-E11.5 period (*Kawasaki et al., 2002*). Thus, ectopic projection phenotypes in these various Sema-Plexin-Nrp mutant backgrounds might be the additive effects of early migration defects coupled with subsequent axon guidance defects. However, *Plxna4* mutants are not compromised in early migration or ganglion extension, and only first show a misprojection phenotype at the time of initial axon extension (*Waimey et al., 2008*).

Our previous (*Manousiouthakis et al., 2014*) and current studies revealed that STG neurons are heterogeneous: there is an Ednra-expressing subset that extends along venous routes to the heart, an Ednra-negative subset that also extends along venous routes to the heart, and an Ednra-negative subset that extends along arterial routes to innervate the neck, limbs, and thoracic body wall. We propose that the Edn1-MEF2C-Plexin-A4 hierarchy described here serves as a basis of second-order axon guidance that controls cardiac sympathetic innervation for the Ednra-expressing subset of STG neurons. We identified *Plxna4* as a downstream target of endothelin signaling, and demonstrated that MEF2C is an endothelin-induced transcription factor that regulates *Plxna4* gene expression in developing neural crest cells. Despite its uniform expression in developing sympathetic neurons (*Cheng et al., 2001*), *Plxna4* gene regulation does not require endothelin-dependent MEF2C transactivation in the Ednra-negative subsets of STG neurons, indicating that *Plxna4* expression is controlled by other gene regulatory mechanism in those sympathetic neurons.

While second-order guidance is sufficient to explain the major component of STG-cardiac circuitry, several further intriguing questions remain. First, an Ednra-negative subset of STG neurons extends along arterial routes. These neurons express Plexin-A4, which might be expected to cause them to avoid their arterial targets. In fact, in a permissive environment as in our collagen explant assay system, all neurites from control STG exhibit strong repulsive response towards Sema3a-overexpressing COS aggregates (*Figure 8*). Presumably, in the in vivo environment, additional mechanisms emanating from arteries override Sema3a-Plexin-A4 signaling. These mechanisms may act as stronger attractive cue(s) for sympathetic axons to cancel the repulsive effects on Semaphorin, or may alter expression and/or post-translational modification of Plexin-A4/Npn receptor signaling components to make arterially-committed axons be non-responsive to arterial Semaphorin. If the latter mechanism exists, this would also be considered as an example of second-order axon guidance. Second, while endothelin signaling is needed for the major subset of cardiac sympathetic nerves to express Plexin-A4 and to avoid arterial targets to reach the heart, some level of cardiac sympathetic innervation from the STG still occurs in endothelin signaling mutants (e.g., *Figure 1f–h*, *Figure 5c–d*). These neurons are repelled from arteries, in a Plexin-A4-dependent, endothelin-independent manner, but these nerves still follow the vena cavae and reach and innervate the SA node and other cardiac targets. These neurons are either responsive to alternative positive, unidentified guidance cues (other than endothelin) that originate in the vena cavae, or simply follow the vena cavae as a neutral default vascular pathway. The presence of these neurons is not sufficient to facilitate a response to amphetamine, suggesting that they might be functionally distinct from the endothelin-dependent subset; alternatively, there may simply be too few of these neurons to have a functional consequence in this specific physiological process. The role of this subset of neurons in heart biology remains unknown.

# Materials and methods

## Key resources table

| Reagent type (species) or resource | Designation | Source or reference | Identifiers | Additional information |
|---|---|---|---|---|
| Genetic reagent (M. musculus) | Ednra$^{fl}$, ETA$^{flox}$ | PMID: 14585980 | MGI:2687362, RRID:IMSR_RBRC06323 | |
| Genetic reagent (M. musculus) | MEF2C$^{fl}$, Mef2c$^{loxP}$ | PMID: 17336904 | MGI:3718323 | |
| Genetic reagent (M. musculus) | Edn1$^-$, EdnRA$^-$, ET-1$^-$ | PMID: 8152482 | MGI:1857637 | |
| Genetic reagent (M. musculus) | Ece1$^-$, ECE-1 - | PMID: 9449665 | MGI:2158947, RRID:IMSR_RBRC06166 | |
| Genetic reagent (M. musculus) | Plxna4$^-$, PLA4 KO, PlexinA4$^-$ | PMID: 15721238 | MGI:3579185, RRID:MMRRC_030407-MU | |
| Genetic reagent (M. musculus) | ThCre, Th::Cre | PMID: 1785595 | MGI:5607431, RRID:MMRRC_029177-UCD | |
| Genetic reagent (M. musculus) | Wnt1Cre, Tg(Wnt1-cre/Esr1)10Rth | PMID: 9843687 | MGI:2447684 | |
| Cell line (Cercopithecus aethiops) | COS-7 | ATCC | CRL-1651, RRID: CVCL_0224 | LOT: 5784927 Originally tested negative for Mycoplasma, Bacterial and fungal contamination (purchase date:6/30/2016). Cells have not independently authenticated for each passage used for experiments and data collection. Cells were not independently tested for mycoplasma contamination. |
| Antibody | Sheep polyclonal anti-Tyrosine Hydroxylase | Millipore | AB1542, RRID:AB_90755 | IHC (1:400) |
| Antibody | Rabbit polyclonal anti-Plexin A4 | Novus Biological | MBP1-85128, RRID:AB_11013411 | IHC (1:400) |
| Antibody | Rabbit polyclonal anti-Semaphorin 3A | Abcam | ab23393, RRID: AB_447408 | IHC (1:300) |
| Antibody | Rabbit polyclonal anti-Semaphorin 3F | Abcam | an203394, RRID:AB_2783521 | IHC (1:500) |
| Antibody | Rabbit polyclonal anti-Semaphorin 6A | Abcam | ab72369 | IHC (1:200) |
| Antibody | Mouse monoclonal anti-Semaphorin 6B | SCBT | sc-390928, RRID:AB_2783522 | IHC (1:200) |
| Antibody | Rabbit polyclonal anti-HCN4 | Abcam | ab69054, RRID: AB_1861080 | IHC (1:500) |
| Antibody | Mouse monoclonal anti-SYP | SCBT | sc-17750, RRID: AB_628311 | IHC (1:500) |
| Recombinant DNA reagent | Ad-GFP-2A-mSema3A | VectorBio | ADV-271580 | |
| Sequence-based reagent | qRT-PCR primers (hprt) | | primer bank ID: 7305155a1 | |
| Sequence-based reagent | qRT-PCR primers (nrp1) | | primer bank ID: 6679134a1 | |

*Continued on next page*

*Continued*

| Reagent type (species) or resource | Designation | Source or reference | Identifiers | Additional information |
|---|---|---|---|---|
| Sequence-based reagent | qRT-PCR primers (plxna3) | | primer bank ID: 667939a1 | |
| Sequence-based reagent | qRT-PCR primers (plxna4) | | primer bank ID: 28461143a1 | |
| Commercial assay or kit | RNaseOUT | Thermofisher | 10777019 | |
| Commercial assay or kit | Trizol LS | Thermofisher | 10296028 | |
| Commercial assay or kit | RNeasy Micro Kit | Qiagen | 74004 | |
| Commercial assay or kit | Ovation RNA-Seq System V2 | NuGEN | | |
| Commercial assay or kit | Encore Rapid Library System | NuGEN | | |
| Chemical compound, drug | amphetamine | Sigma A5880 | | |
| Chemical compound, drug | isoproterenol | Sigma I6504 | | |
| Chemical compound, drug | propranolol | Sigma P0884 | | |
| Software, algorithm | Fiji - Image J | https://imagej.net/Fiji | RRID:SCR_003070 | |
| Software, algorithm | CASAVA (v1.8.2) | Illumina | RRID:SCR_001802 | |
| Software, algorithm | tophat v1.3 | *Trapnell et al. (2009)* | doi:10.1093/bioinformatics/btp120 | |
| Software, algorithm | U-Seq version 8.0.2 | *Nix et al. (2008)* | doi:10.1186/1471-2105-9-523 | |
| Software, algorithm | LabChart 8 | AD Instruments | RRID:SCR_001620 | |

## Animals

The *Th-Cre* (*Gong et al., 2007*), *Wnt1-Cre* (*Danielian et al., 1998*), *Ednra* (*Clouthier et al., 1998*), *Edn1* (*Kurihara et al., 1995*), *Ece1* (*Yanagisawa et al., 1998*), *Plxna4* (*Yaron et al., 2005*) (MMRRC U420D020918), conditional *Ednra* (*Kedzierski et al., 2003*), and conditional *Mef2c* (*Arnold et al., 2007*) alleles have been described previously. All experiments with animals complied with National Institute of Health guidelines and were reviewed and approved by the CHLA (274-18), UCSF (AN171342) or MUSC (2018–00627) Institutional Animal Care and Use Committee.

## Wholemount Th immunostaining

Mouse embryo thoraxes were dissected from PFA fixed embryos, dehydrated through a methanol series (50–100%), and incubated overnight in 3%$H_2O_2$/20%DMSO/methanol solution to quench endogenous peroxidase activity. Tissues were rehydrated through a methanol series (100–50%) to PBST (1% Tween-20 in PBS), and incubated with sheep anti-Th antibody (1:400; Millipore) for 3 days at 4°C, followed by HRP-conjugated secondary antibody for 3 days at 4°C. Immunoreactive signal was visualized by DAB detection. Tissues were then dehydrated through a methanol series and cleared with benzyl benzoate/benzyl alcohol (2:1). Postnatal hearts were dissected from animals fixed with 4% PFA in PBS by retroaortic perfusion through the dorsal aorta, and then processed for immunostaining as described above. To quantify Th$^+$ axons in the embryonic thoraxes and postnatal hearts, the Th labeled area (binary threshold selection) and the total region of interest (dorsal body wall of the thorax or dorsal surface of the heart) pixel area were measured using Fiji-Image J software.

## Immunohistochemistry and immunofluorescence

PFA fixed embryos were dehydrated, cleared and embedded in paraffin. 10 µm paraffin sections were dewaxed, rehydrated in ethanol series (100–70%) to $H_2O$. Following antigen retrieval by

heating in citrate buffer (0.1 M sodium citrate, pH6.0), sections were immunolabeled with anti-Th (1:400; Millipore AB1542), anti-Plexin-A4 (1:400; Novus NBP 1–85128), anti-Sema3a (1:300; Abcam ab23393), anti-Sema3f (1:500; Abcam ab203394), anti-Sema6a (1:200; Abcam ab72369) or anti-Sema6b (1:200; SCBT sc-390928) antibodies. For immunohistochemistry, sections were then incubated with HRP-conjugated secondary antibodies (1:200). Immunoreactive signal was visualized by DAB detection system followed by counterstaining with hematoxylin. For immunofluorescence labeling, histological sections were treated with 0.03% Sudan Black to quench autofluorescence prior to primary antibody incubation. Immunoreactive signals were visualized using Alexa Fluor-conjugated secondary antibodies (Life Science) and counterstained with DAPI. Plexin-A4 immunoreactive signal was visualized with biotin-streptavidin amplification.

For postnatal hearts, 300 µm thick vibratome sections were prepared for immunofluorescence staining. Sections were immunolabeled with anti-HCN4 (1:500; Abcam ab69054), anti-Th (1:400: Millipore AB1542) and anti-SYP (1:500; SCBT sc-17750) antibodies overnight at 37°C, and followed by Alexa Fluor-conjugated secondary antibodies (Life Science) for 3 hr at 37°C. Immunolabeled sections were counterstained with DAPI, cleared with Sca*l*eU2 and flat-mount to histology slides for confocal microscope image acquisition. To quantify SYP$^+$ area in Th$^+$ axons that innervate the SA node, the HCN4$^+$ pixel area was measured, and the SYP$^+$ and Th$^+$ areas were measured by binary threshold selection using Fiji-Image J software.

## Embryo explant culture, Edn1-treatment, and RNA-sequencing

Embryos were explanted at E9.5, and cultured as described previously (*Rojas et al., 2005*). Explants were cultured for 1 hr prior to treatment with 10 µM Edn1 or vehicle for 24 hr. Following treatment, three explants were pooled for each sample and digested in 300 µl 0.25% trypsin, 0.7 mM EDTA in PBS for 30 min at 37°C. A single cell suspension was formed and 30 µl fetal bovine serum, 10 µl 1M HEPES [pH7.4], and 10 µl RNase inhibitor (RNaseOUT, Invitrogen) were added. Cells were FACS sorted into 500 µl of Trizol LS (Invitrogen) and RNA was prepared using the RNeasy Micro kit (Qiagen). cDNA was prepared and amplified from 2 ng of total RNA using the Ovation RNA-Seq System V2 (NuGEN). Amplified cDNA was fragmented to an average of 200–300 bp before ligating to barcoded-TruSeq adaptors using the Encore Rapid Library System (NuGEN). Libraries were pooled and sequenced for $>5 \times 10^7$ reads on a HiSeq 2000 (Illumina). Fastq files were obtained after demultiplexing with CASAVA (v1.8.2) and aligned to the reference mouse genome (mm9) using tophat v1.3 (*Trapnell et al., 2009*). Differential expression was determined with U-Seq version 8.0.2 (*Nix et al., 2008*).

## Bioinformatics

Previously published ChIP-seq datasets available as BigWig files (*Telese et al., 2015*) were uploaded and visualized on the UCSC Genome Browser. For comparative sequence analyses of evolutionary conservation in the *Plxna4* locus and identification of conserved MEF2 sites, the Evolutionary Conserved Regions (ECR) Browser suite of tools was used (https://rvista.dcode.org) (*Frazer et al., 2004*).

## Collagen gel explant culture

E14.5 STG were dissected in PBS, cut into several pieces and placed into a rat-tail-collagen gel as described previously (*Manousiouthakis et al., 2014*). For STG-dorsal aorta co-culture, E14.5 STG and thoracic aorta segments were separately dissected from *Ednra$^{-/-}$*, *Th-Cre/Ednra$^{fl/-}$* and their littermate controls. Control, *Ednra$^{-/-}$*, *Th-Cre/Ednra$^{fl/-}$* STG were placed in a collagen gel at the proximity of the thoracic aorta segments of the indicated genotype and grown in L15-CO$_2$ medium supplemented with 10% FBS for 3 days, then fixed with 4% PFA. Neurite outgrowth was evaluated by immunostaining the explants with anti-Th (1:400; Millipore AB1542) antibody followed by detection using Alexa Fluor secondary antibodies (Life Science). To quantify neurite outgrowth, the fluorescent labeled neurites extending from explants were measured by binary threshold selection using Fiji-Image J software.

For STG-COS aggregate co-culture, COS7 (ATCC CRL-1651, LOT: 5784927) cell aggregates were prepared in 40 µl hanging drops of $5 \times 10^5$ cells/ml suspension of untransfected or Ad-GFP-2A-mSema3A (VectorBio ADV-271580) transfected COS cells for 3 days at 37°C. Separately, E14.5 STGs

were dissected from *Ednra*<sup>-/-</sup>, *Plxna4*<sup>-/-</sup> and their littermate controls. Control, *Ednra*<sup>-/-</sup>, *Plxna4*<sup>-/-</sup> STGs were placed in a collagen gel at the proximity of untransfected-COS cell aggregates or between untransfected- and Sema3a-transfected-COS cell aggregates, and grown in L15-$CO_2$ medium supplemented with 10% FBS for 4 days. Neurite outgrowth was evaluated as described above.

## Quantitative real-time RT-PCR

Single STGs were dissected individually from E14.5 *Ednra*, *Edn1* and *Ece1* mutants and their littermate control embryos. Total RNA was extracted from individual STG by guanidinium isothiocyanate extraction and reverse transcribed using M-MLV reverse transcriptase. All samples were analyzed independently by real-time PCR using iCycler iQ with SYBR green supermix (Bio-Rad) with the following primer pairs; *hprt* (primer bank ID: 7305155a1, 5'-TCAGTCAACGGGGGACATAAA-3', 5'-GGGGCTGTACTGCTTAACCAG-3'), *nrp1* (primer bank ID: 6679134a1, 5'-GACAAATG TGGCGGGACCATA-3', 5'-TGGATTAGCCATTCACACTTCTC-3'), *plxna3* (primer bank ID: 667939a1, 5'-CAGATACCACTCTGACTCACCT-3', 5'-GGCCCGTAGCTCAGTTAGG-3'), *plxna4* (primer bank ID: 28461143a1, 5'-ACAGGGCACATTTATTTGGGG-3', 5'-CACTTGGGGTTGTCCTCATCT-3').

## Electrocardiography (ECG)

6–7 week-old mice were anesthetized by isoflurane inhalation via a SomnoSuite small animal anesthesia system (Kent Scientific), and needle electrodes were inserted subcutaneously in a standard lead II arrangement (at the right thoracic limb and the left pelvic limb) to obtain ECG readings with a Powerlab 35 data acquisition system (AD Instruments). Once stable baseline heart rate was confirmed, drug was administered by intraperitoneal injection. Dosage of drug used was 2 mg/kg for amphetamine, 10 µg/kg for isoproterenol, and 25 µg/kg for propranolol. The ECG for each subject was recorded continuously for the entire period of the experiment. LabChart eight software (AD Instruments) was used for heart rate and heart rate variability analyses.

## Statistics

All quantified data were graphed as mean ± s.e.m. or mean ± s.d., and analyzed for significance using an unpaired Student's t-test. Minimum sample sizes were determined based on population size, confidence level and standard errors/deviations.

## Acknowledgements

We thank Catherine Bridges and Chris Cowan for provision of additional *Mef2c* mutant embryos to complete this study. This research was supported by grants NS062901 and NS083265 from National Institutes of Health to TM, and by grants HL064658 and HL136182 from National Institutes of Health to BLB.

## Additional information

### Funding

| Funder | Grant reference number | Author |
| --- | --- | --- |
| National Institute of Neurological Disorders and Stroke | NS062901 | Takako Makita |
| National Institute of Neurological Disorders and Stroke | NS083265 | Takako Makita |
| National Heart, Lung, and Blood Institute | HL064658 | Brian L Black |
| National Heart, Lung, and Blood Institute | HL136182 | Brian L Black |

The funders had no role in study design, data collection and interpretation, or the decision to submit the work for publication.

## Author contributions
Denise M Poltavski, Data curation, Formal analysis, Validation, Investigation, Visualization, Methodology, Writing—original draft, Writing—review and editing; Pauline Colombier, Resources, Data curation, Software, Formal analysis, Validation, Investigation, Methodology, Writing—review and editing; Jianxin Hu, Data curation, Formal analysis; Alicia Duron, Data curation, Validation; Brian L Black, Resources, Data curation, Software, Formal analysis, Supervision, Funding acquisition, Validation, Investigation, Visualization, Methodology, Writing—review and editing; Takako Makita, Conceptualization, Resources, Data curation, Software, Formal analysis, Supervision, Funding acquisition, Validation, Investigation, Visualization, Methodology, Writing—original draft, Project administration, Writing—review and editing

## Author ORCIDs
Brian L Black  http://orcid.org/0000-0002-6664-8913
Takako Makita  http://orcid.org/0000-0002-5598-5690

## Ethics
Animal experimentation: All experiments with animals complied with National Institute of Health guidelines and were reviewed and approved by the Children's Hospital Los Angeles (274-18), UCSF (AN171342) or MUSC (2018-00627) Institutional Animal Care and Use Committee.

## Decision letter and Author response
Decision letter https://doi.org/10.7554/eLife.42528.018
Author response https://doi.org/10.7554/eLife.42528.019

# Additional files

## Supplementary files
• Transparent reporting form
DOI: https://doi.org/10.7554/eLife.42528.014

## Data availability
Previously published ChIP-seq datasets available as BigWig files (Telese et al. 2015) were uploaded and visualized on the UCSC Genome Browser.

The following previously published dataset was used:

| Author(s) | Year | Dataset title | Dataset URL | Database and Identifier |
|---|---|---|---|---|
| Telese F, Ma Q, Perez PM, Notani D, Oh S, Li W, Comoletti D | 2015 | LRP8-Reelin-Regulated Neuronal Enhancer Signature Underlying Learning and Memory Formation | https://www.ncbi.nlm.nih.gov/geo/query/acc.cgi?acc=GSE66710 | NCBI Gene Expression Omnibus, GSE66710 |

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
