## [Decision Letter]

Thank you for submitting your article "Venous endothelin modulates responsiveness of cardiac sympathetic axons to arterial semaphorin" for consideration by *eLife*. Your article has been reviewed by Didier Stainier as the Senior Editor, a Reviewing Editor, and three reviewers. The reviewers have opted to remain anonymous.

The reviewers have discussed the reviews with one another and the Reviewing Editor has drafted this decision to help you prepare a revised submission.

Summary:

In the present study, Poltavski et al., demonstrate a critical role for endothelin in the stellate ganglion sympathetic neuron axonal guidance, including its role in upregulation of plexina4 via MEF2c which is required to prevent aberrant projections. This work displays several exciting new findings regarding axon guidance, with particular focus on the actions of Edn1. This endothelin plays a role in instructing axon guidance toward veins and inducing repulsion from incorrect arterial routes. The findings provide a significant contribution for understanding how axons arrive at their proper targets by suggesting a mechanism by which axon attraction and repulsion work synergistically with a hierarchal system of axon guidance cues. This manuscript is complete, extensive, and elegantly designed and contributes to a nuanced view of axonal guidance.

Essential revisions:

1) The "genetic interaction" experiments in Figure 5 are difficult to interpret because plexA4 and Ednra heterozygous controls for the histological analysis are lacking. These should be included.

2) Figure 8 is not described in the Results section, but it is mentioned in the Discussion. All findings should be first described in the Results section of the paper.

3) The Mef2cNCKO analysis – based on the expression analysis, one would expect a guidance phenotype as presented in Figure 1. The authors say " Just as observed in Ednra mutants (Figure 3B-M), axons associated with the superior vena cavae retained Plexin-A4 expression in Mef2cNCKO mutant embryos (Figure 4D, G-H). In contrast, sympathetic axons associated with the thoracic aorta did not express Plexin-A4 in Mef2cNCKO mutant embryos (Figure 4D, I-J), which we also observed in Ednra mutants (Figure 3C, J-M)". Are these misprojecting axons? Clearer images are needed to make this more convincing.

4) The paper should include a model in the last Figure.

---

## [Author Response]

Essential revisions:1) The "genetic interaction" experiments in Figure 5 are difficult to interpret because plexA4 and Ednra heterozygous controls for the histological analysis are lacking. These should be included.

We used *Ednra* heterozygote (*ThCre; Ednra^fl/+^*) as the original control in Figure 5. We have now incorporated Plxna4 heterozygote (*Plxna4^+/-^*) into the revised Figure 5 as an additional control for proper interpretation of the results. This is the second vertical column of the revised figure, showing the whole mount heart and sections below it, under the heading *Plxna4^+/-^*.

2) Figure 8 is not described in the Results section, but it is mentioned in the Discussion. All findings should be first described in the Results section of the paper.

We have moved the text associated with the previous Figure 8 into Results section, which is now renumbered as Figure 7 in the revised manuscript (because of its new position in the text). The previous Figure 7 is now Figure 8.

3) The Mef2cNCKO analysis – based on the expression analysis, one would expect a guidance phenotype as presented in Figure 1. The authors say " Just as observed in Ednra mutants (Figure 3B-M), axons associated with the superior vena cavae retained Plexin-A4 expression in Mef2cNCKO mutant embryos (Figure 4D, G-H). In contrast, sympathetic axons associated with the thoracic aorta did not express Plexin-A4 in Mef2cNCKO mutant embryos (Figure 4D, I-J), which we also observed in Ednra mutants (Figure 3C, J-M)". Are these misprojecting axons? Clearer images are needed to make this more convincing.

We agree with the reviewer that this is an important demonstration, and have now included a wholemount view of the misprojection phenotype in the *Mef2c* mutant background, shown as the new Figure 4C-D. We feel that the phenotypic similarity of this mutant to the *Ednra* and *Ednra/Plxna4* mutants is compelling.

4) The paper should include a model in the last Figure.

We have included a model diagram as the last Figure (Figure 9).